# Elaboration and Characterization of *Pereskia aculeate* Miller Extracts Obtained from Multiple Ultrasound-Assisted Extraction Conditions

**DOI:** 10.3390/metabo13060691

**Published:** 2023-05-26

**Authors:** Maria Clara Coutinho Macedo, Viviane Dias Medeiros Silva, Mateus Sá Magalhães Serafim, Vinícius Tadeu da Veiga Correia, Débora Tamires Vitor Pereira, Patrícia Regina Amante, Antônio Soares Júnior da Silva, Henrique de Oliveira Prata Mendonça, Rodinei Augusti, Ana Cardoso Clemente Filha Ferreira de Paula, Júlio Onésio Ferreira Melo, Christiano Vieira Pires, Camila Argenta Fante

**Affiliations:** 1Departamento de Alimentos, Faculdade de Farmácia, Campus Belo Horizonte, Universidade Federal de Minas Gerais, Belo Horizonte 31270-901, Brazil; 2Departamento de Ciências Exatas e Biológicas, Campus Sete Lagoas, Universidade Federal de São João del-Rei, Sete Lagoas 35702-031, Brazil; 3Departamento de Análises Clínicas e Toxicológicas, Faculdade de Farmácia, Campus Belo Horizonte, Universidade Federal de Minas Gerais, Belo Horizonte 31270-901, Brazil; 4Departamento de Engenharia e Tecnologia de Alimentos, Faculdade de Engenharia de Alimentos, Universidade Estadual de Campinas, Campinas 130862-862, Brazil; 5Universidade Federal do Rio Grande do Norte, Distrito de Macaíba 59280-000, Brazil; 6Departamento de Pedagogia, Instituto de Educação Ciência e Tecnologia do Paraná, Palmas 85555-000, Brazil; 7Departamento de Química, Universidade Federal de Minas Gerais, Belo Horizonte 31270-901, Brazil; 8Departamento de Ciências Agrárias, Campus Bambuí, Instituto Federal de Educação, Ciência e Tecnologia de Minas Gerais, Bambuí 38900-000, Brazil; 9Departamento de Engenharia de Alimentos, Campus Sete Lagoas, Universidade Federal de São João del-Rei, Sete Lagoas 35702-031, Brazil

**Keywords:** paper spray mass spectrometry, infrared spectroscopy, *P. aculeate*, PANC, green technology

## Abstract

*Pereskia aculeata* Miller, is an unconventional food plant native to South America. This study aimed to investigate the influence of different ultrasonic extraction times (10, 20, 30, and 40 min) on the phytochemical profile, antioxidant and antibacterial activities of ethanolic extracts obtained from lyophilized *Pereskia aculeate* Miller (ora-pro-nobis) leaves, an under-researched plant. Morphological structure and chemical group evaluations were also conducted for the lyophilized *P. aculeate* leaves. The different extraction times resulted in distinct phenolic content and Antioxidant Activity (ATT) values. Different extraction time conditions resulted in phenolic compound contents ranging from 2.07 to 2.60 mg EAG.g^−1^ of extract and different ATT values. The ATT evaluated by DPPH was significantly higher (from 61.20 to 70.20 μM of TE.g^−1^ of extract) in extraction times of 30 and 40 min, respectively. For ABTS, it varied between 6.38 and 10.24 μM of TE.g^−1^ of extract and 24.34 and 32.12 μM ferrous sulp.g^−1^ of extract. All of the obtained extracts inhibited the growth of *Staphylococcus aureus*, particularly the treatment employing 20 min of extraction at the highest dilution (1.56 mg.mL^−1^). Although liquid chromatography analyses showed that chlorogenic acid was the primary compound detected for all extracts, Paper Spray Mass Spectrometry (PS-MS) suggested the extracts contained 53 substances, such as organic, fatty, and phenolic acids, sugars, flavonoids, terpenes, phytosterols, and other components. The PS-MS proved to be a valuable technique to obtain the *P. aculeate* leaves extract chemical profile. It was observed that the freeze-drying process enhanced the conservation of morphological structures of *P. aculeate* leaves, as evidenced by scanning electron microscopy (SEM). Fourier transform infrared spectroscopy (FTIR) identified carboxyl functional groups and proteins between the 1000 and 1500 cm^−1^ bands in the *P. aculeate* leaves, thus favoring water interaction and contributing to gel formation. To the best of our knowledge, this is the first study to evaluate different times (10, 20, 30 and 40 min) for ultrasound extraction of *P. aculeate* leaves. The polyphenols improved extraction, and high antioxidant activity demonstrates the potential for applying *P. aculeate* leaves and their extract as functional ingredients or additives in the food and pharmaceutical industries.

## 1. Introduction

Plant materials are natural sources of phytochemicals. In the human body, these compounds perform crucial roles such as protection, antioxidant, anti-inflammatory, antifungal, and anti-bacterial activities. On top of that, phytochemicals act in the immune system stimulation [1,2]. A *Pereskia aculeate* Miller, known worldwide as ora-pro-nobis, is a representative of phytochemical-rich plants. *P. aculeate* is a South American plant species, known by the acronym UFP(Unconventional Food Plant) (PANC, in Portuguese), belonging to the Cactaceae family and Pereskioideae subfamily. The plant’s leaves, its non-toxic edible portion, are used in traditional medicine and cooking [1,2].

The *P. aculeate* leaves are a nutritional complement primarily because of their high protein, fiber, calcium, and iron contents. Furthermore, *P. aculeate* extracts and leaves were also found to be rich in carotenoids and phenolic compounds, valuable phytochemicals which provide healing, anti-inflammatory, antifungal, and antioxidant activities, as well as analgesic potential [3,4,5,6,7].

While Cruz et al. [8] reported significant antioxidant and anti-hemolytic activities for ora-pro-nobis leaves, Torres et al. [9] identified anti-inflammatory and anticholinergics activities. The extracts obtained by Massocatto et al. [10] from *P. aculeate* leaves and fruit presented anticholinesterasic, cytotoxic, and antiproliferative effects. Pinto et al. [11,12] observed that ora-pro-nobis methanolic extract and hexanic fractios containing gels enhanced the excisional wound healing in mice, as well as analgesic effect of hydromethanolic fractions. These publications indicated the *P. aculeate* extract’s potential for applications in the pharmaceutical industry. According to Souza et al. [7], the essential oil obtained from *P. aculeate* leaves presented antioxidant, antimicrobial, and antifungical activities in in vitro assays. Considering this vast literature, one may infer that the ora-pro-nobis has a considerable potential to be used as a valuable raw-material, ingredient, or additive in the food/pharmaceutical industries.

Currently, the consumption of nutraceuticals and food supplements has increased, as the consumers have become more aware of the benefits of consuming products rich in bioactive compounds, which represent health benefits to the human body (e.g., disease prevention) [13]. Bioactive compounds with antioxidant capacity can be found in complex vegetable matrices. Hence, to obtain such compounds, it is necessary to employ more efficient and selective extraction techniques to the analyte of interest.

The most known traditional extraction methods are maceration, Soxhlet, and mechanical agitation. These methods are based on high temperature, long process times, and high solvent consuming techniques [14]. Moreover, in general, the solvents used in these methods are toxic and harmful to health and the environment.

Specific extraction conditions such as temperature, may directly affect the antioxidant and biological potential of compounds obtained from natural extracts [15]. Thus, more efficient extraction techniques providing satisfactory yields at mild temperatures such as the ultrasound-assisted extraction are promising alternatives. Compared to conventional methods for bioactive compound extraction, Ultrasound-Assisted Extraction (UAE) offers significant benefits regarding extraction time and process security [16,17].

The UAE technique is based on the formation of mechanical waves that form small bubbles. These bubbles grow and implode, resulting in the cavitation phenomenon. Then, the disruption of the vegetable material cell wall takes place, increasing its surface contact and allowing the entrance of the solvent in the matrix, consequently intensifying the mass transfer. UAE is a rapid technique which produces extracts with high yields. Furthermore, it is considered to be an environmentally friendly method because of its shorter extraction time and solvent consumption compared to conventional techniques [15,16]. Furthermore, the UAE enhances the phytomolecules yields and extracts/isolated biological efficiency [16,18].

Recent studies have shown that UAE is efficient for recovering different compounds (e.g., anthocyanins, phenolic compounds, flavonoids, carotenoids, and antioxidants) from various vegetal matrices [15,17]. Nevertheless, we could not find publications using UAE to obtain bioactive compounds from *P. aculeate* leaves and there are just few publications in the literature identifying chemical compounds found in *P. aculeate* leaves [19]. Moreover, it is crucial to understand the impact of extraction techniques on the proportion of bioactive compounds found in the extracts and leaves. It is also important to mention that natural extracts such as those obtained from ora-pro-nobis leaves may act as food preservatives, besides their health-beneficial effects, when applied in food production. These extracts can be substitutes for broadly used synthetic preservatives in food products [20,21].

In this context, the present paper aims to investigate the influence of different ultrasonic extraction times on the phytochemical profile and antioxidant and antibacterial activities of ethanolic extracts obtained from lyophilized *P. aculeate* leaves. In addition, a characterization of the morphological structure and presence of chemical groups was conducted. This study is intended to contribute to the scientific advance of the area by allowing the obtaining of optimized ultrasound-assisted extraction time of bioactive compounds form ora-pro-nobis leaves. The obtained extracts may be used as potential additives in food and pharmaceutical products.

## 2. Materials and Methods

### 2.1. Chemical Reagents and Bacterial Strains

The Folin–Ciocalteu, 2,2′-azino-bis (3-ethylbenzothiazoline-6-sulfonic acid) diamonic salt (ABTS), 2,4,6-Tris(2-pyridyl)-s-triazine (TPTZ), 2,2-diphenyl-1-picrylhydrazyl (DPPH), 6-Hydroxy-2,5,7,8-tetramethylchroman-2-carboxylic acid (Trolox), catechin, caffeic acid, ellagic acid, quercetin, and chlorogenic acid reagents were supplied by from Sigma-Aldrich (San Luis, MO, USA,). All reagents were of analytical grade.

The reference strains of the *Staphylococcus aureus* ATCC 29213, *Escherichia coli* (ATCC 35218; beta-lactamase producer) and methicillin resistant *Staphylococcus aureus Staphylococcus aureus* resistant to methicillin (MRSA; ATCC 43300) bacteria were gently provided by the Clinical Microbiology Laboratory from the Faculty of Pharmacy at UFMG.

### 2.2. Plant Sampling

The *P. aculeate* leaves were collected from a crop at the Federal University of São João del-Rei (UFSJ, Sete Lagoas, Brazil) in March 2019.

### 2.3. Lyophilization

The *P. aculeate* leaves were selected according to their visual appearance (typical green color and integrity). After selection, the leaves were sanitized with flowing water and dried with paper towels. Thus, the material was submitted to freeze in an Ultra Low-Temperature Freezer at −55 °C for 24 h.

The lyophilization was carried out by a lyophilizer (LS-6000-A model, Terroni Equipamentos Científicos, São Carlos, Brazil) for 12 h. The leaves were conditioned in plastic bags at −18 °C until further analyses when the material was unfrozen, crushed in a blender (Philips Walita, São Paulo, Brazil), and sifted in a 16-mesh sieve.

### 2.4. Extracts Preparation

For the extract preparation, 5 g of the sample was mixed with 50 mL of a 50% ethyl alcohol solution. The mixture was weighed in an Erlenmeyer wrapped with aluminum foil and covered with plastic paraffin film (Parafilm®, São Paulo, Brazil) and aluminum foil to avoid alcohol evaporation during extraction.

The extracts were prepared using a digital ultrasonic washer (SoniClean 2, Sanders Medical, Santa Rita do Sapucaí, Brazil) at room temperature (20 °C), frequency 42 kHz, and power 160 W. Four different extraction times were evaluated, namely, 10, 20, 30, and 40 min.

After extractions, filtration was carried out with qualitative filter paper. The obtained extracts were vaporized in a rotary evaporator (R-215, Büchi, Valinhos, Brazil) at 45 °C until the alcohol removal, resulting in aqueous extracts, named EO10, EO20, EO30 and EO40 corresponding to 10, 20, 30 and 40 min ultrasound treatment, respectively.

### 2.5. Total Phenolic Compounds and Antioxidant Activity

Following the Folin–Ciocalteu method [22], the total phenolic compounds content was determined, with results expressed in equivalents of gallic acid (mg EAG.g^−1^ of the extract). The Antioxidant Activity (ATT) was evaluated using three distinct methodologies: (1) reaction with 2,2-diphenyl-1-picrylhydrazyl (DPPH) [23]; (2) free radical capture 2,2′-azino-bis (3-ethylbenzothiazoline-6-sulfonic acid) (ABTS) [24]; (3) Ferric Reducing Antioxidant Power (FRAP) [24]. The analyses were carried out in a dark room and triplicate.

### 2.6. Phenolic Compounds Identification and Quantification

High-Performance Liquid Chromatography (HPLC) analyses were conducted according to the method described by Eça et al. [25]. The four extracts were filtered in a 0.22 μm nylon filter syringe and injected in HPLC equipment (Acquity UPLC® Class, Milford, Waters, MA, USA), equipped with a UV detector by diode array, quarternary bomb, degasser, autosampler, and column (Acquity UPLC ® BEH C18—2.1 × 100 mm; 1.7 µm, Waters, Milford, MA, USA). The flow was kept constant at 0.3 mL.min^−1^, with two mobile phases (A = acetonitrile and B = water : formic acid, 99.75:0.25).

Calibration curves used to identify the compounds were prepared with the following external standards: gallic acid, catechin, and chlorogenic acid (diluted in water for complete dissolution); caffeic acid, ellagic acid, and quercetin (diluted in methanol for complete dissolution). While the water diluted standards were run for 10 min in an isocratic elution mode of 5:95 *v/v* (A : B), a linear gradient was applied to the other standards: 0 min, 8% (A); 8 min, 15% (A); 14 min, 25% (A); 20 min, 8% (A). The UV spectra were obtained between 200 and 400 nm. Chromatograms were processed at 253, 271, 320, and 372 nm. The calibration curves were built within the following ranges: gallic acid and quercetin (21.2 to 212 μg.g^−1^), catechin (24.4 to 244 μg.g^−1^), chlorogenic acid (18.4 to 184 μg.g^−1^), and ellagic acid and caffeic acid (20 to 200 μg.g^−1^). All data acquired from these analyses were processed in the Empower® software, and the results were expressed in µg.g^−1^ of extract.

### 2.7. Antibacterial Activity

Antibacterial activity of the four extracts (E10, E20, E30 and E40) was assessed by the broth microdilution method in 96-well microplates, according to the Clinical and Laboratory Standards Institute [26]. First, 200 µL of each extract was added to 100 µL of Mueller Hinton broth (MHB). Serial dilutions (100 µL) were made in order to reach concentrations from 50 to 1.56%, that is, 50 to 1.56 mg.mL^−1^, after adding each bacterial suspension (100 μL) containing 1.5 × 10^5^ colony forming units per milliliter (CFU.mL^−1^). A growth inhibition controls (0.1 µg.mL^−1^ of penicillin for *S. aureus*, 2.5 µg.mL^−1^ of gentamicin for *E. coli* and 2.5 µg.mL^−1^ of streptomycin for MRSA), a cell viability control (bacteria only), and a sterility control (MHB only) were used in triplicate in each assay. After incubation at 37 °C for 24 h, microplates were inspected visually for inhibition of bacterial growth, and wells with no visible growth were considered as the minimal inhibitory concentration (MIC), that is, the lowest concentration (mg.mL^−1^) of the extracts in which no visible bacterial growth was evidenced. All conditions were tested in triplicate in at least two independent assays.

### 2.8. Fourier-Transform Infrared (FTIR) Spectroscopy

The lyophilized *P. aculeate* leaves spectrum was acquired in the 4000 range at 500 cm^−1,^ with 20 scannings by experiment (resolution of 4 cm^−1^). The FTIR spectroscopy was performed in a spectrometer (Shimadzu IRAffinity—1) with DLaTGS detector (Deuterated Triglycine Sulfate and Doped with L-alanine) and total attenuated reflectance accessory (ATR) with zinc crystal.

### 2.9. Scanning Electron Microscopy (SEM)

The microstructure of the lyophilized *P. aculeate* leaves fragments was evaluated using a scanning electron microscope with X-rays dispersive energy detector (Leo 440i, EDS: 6070, LEO Electron Microscopy/Oxford, Cambridge, UK). Before the analyses, the lyophilized fragments were dried and kept for 24 h in a vacuum. Subsequently, they were coated with a gold layer of 200 A° in a Sputter Coater EMITECH spray applicator (Model: K450, Kent, UK). The sample surfaces were assessed under vacuum, using an acceleration tension of 10 kV and 500×, 800×, 2000×, and 3000× magnification.

### 2.10. Chemical Profile Determination by Paper Spray Mass Spectrometry (PS—MS)

The extract presenting the best results for antioxidant and antimicrobial activities was evaluated in terms of chemical profile using an LCQ Fleet mass spectrometer (Thermo Scientific, Waltham, MA, USA), equipped with a paper spray ionization source, according to the methodology reported by Silva et al. [27]. The positive and negative ionization modes were used in triplicate. The equipment was used with source voltage equal to +5.0 kV (positive ionization mode) and −3.0 kV (negative ionization mode); capilar voltage of 4.0 V; transference tube temperature of 275 °C; tube lens voltage of 115 V; mass range from 100 to 1000 *m/z*.

An equilateral triangle-shaped chromatography paper (1.5 cm) was positioned in front of the mass spectrometer entrance to carry out the analyses. The paper was supported by a metallic connection and placed at a distance of 0.5 cm with a mobile platform (XYZ). This device was connected to a high-tension mass spectrometer source by a copper wire. Then, 2.0 μL of the extract was applied to the tip of the equilateral triangle-shaped chromatography paper, 40 μL of methanol was transferred to the paper, and the voltage source was switched on for data acquisition. The *m/z* was compared with the literature data and the subsequent fragmentation by sequential mass spectrometry for compound identification. Collision energies for compound fragmentation varied from 15 to 30 eV. The obtained mass spectra were processed with the Xcalibur software (Thermo Scientific, Waltham, MA, USA).

### 2.11. Statistical Analysis

The results were expressed with the replica means and their respective standard deviations. Results normalization, variance homogeneity, analysis of variance, and the mean comparison by Tukey test at 5% were carried out by the SPSS 15.0 program (SPSS Inc., Chicago, IL, USA).

## 3. Results and Discussion

### 3.1. Total Phenolic Compounds and Antioxidant Activity

Table 1 shows the *P. aculeate* leaves total phenolic compounds content and antioxidant activities.

It is essential to mention that the protocols employed in the antioxidant activity assessment were chosen because of their complementary features. Hence, it was possible to identify a multitude of bioactive compounds performing this activity in the samples.

In this study, the total phenolic compounds contents ranged from 2.07 to 2.60 mg EAG.g^−1^ of extract. Overall, it was possible to observe that different extraction times led to different phenolic content and antioxidant activity values. On the other hand, Sim et al. [28] examined the phenolic content of *Pereskia grandifolia* leaves extracts. The authors reported results of 19.08 mg EAG.g^−1^ using hexane as a solvent. Souza et al. [7], who also evaluated *P. aculeate* leaves chemical composition and biological activity, detected 5.17 mg EAG.g^−1^, 11.78 mg EAG.g^−1^, and 15.04 mg EAG.g^−1^ of phenolic compounds in the extracts, using chloroform, petroleum ether, and methanol as solvents. Additionally, different technologies of pressurized fluids were used by Torres et al. [9] to obtain antioxidante-rich extracts from *P. aculeate* leaves. The highest yields were obtained using Soxhlet extraction with ethanol (38.18 mg EAG.g^−1^) and Pressurized Liquid Extraction (PLE) (60.09 mg EAG.g^−1^) using ethanol as solvent at 110 °C. Cruz et al. [8] achieved high total phenolic values (64–65 mg EAG.g^−1^) using mechanical agitation conventional extraction and various mixture proportions of water, acetone, and ethanol.

Those variations of total phenolic compounds might result from diverse factors such as climate, cultivar, maturation stage and plant genetics. Moreover, the samples preparation techniques such as drying methods and type/condition of extraction also affect the recovery yield of phenolic compounds [14,15]. Solvents such as hexane, chloroform, petroleum ether, and methanol provide high extraction yields, but are toxic and restricted to food applications, as well as generate environmentally dangerous waste.

In the extraction aided by ultrasound, the bubbles formed during the process are responsible for the matrix cavitation. When in contact with the vegetal cell structure, they can disrupt the polymers by mechanical effect, creating microspores and allowing the extraction of polyphenols [29,30]. Therefore, it can be inferred that shorter ultrasound extraction times are also effective for disrupting the vegetal membrane, thus, releasing bioactive compounds.

*P. aculeate* leaves antioxidant activities are assigned mainly to secondary metabolites and some primary metabolites, particularly phenolic compounds and flavonoids [31]. Table 1 shows that AAT by DPPH was significantly higher for longer extraction times. This result may be explained by the higher extraction of bioactive compounds that have the ability to scavenge free radicals (DPPH), which was not possible in the shorter exposure times for this same sample. Souza et al. [7], when evaluating the *P. aculeate* leaves chemical composition and biological activity, also found higher antioxidant activity using DPPH in the extracts presenting higher phenolic compound content, similar to extract EO40. Cruz et al. [8] demonstrated that mixtures between water + ethanol and water + acetone provided high antioxidant capacities, measured by FRAP and DPPH. Torres et al. [9,32] evaluated PLE with pure water and ethanol at 80 °C, reporting antioxidant capacity of 0.25 mmol of TE.g^−1^ of extract (FRAP) and 1315 µmol of TE.g^−1^ of extract (ABTS), respectively. The authors reported that the highest ora-pro-nobis extracts antioxidant activities, measured with FRAP, were obtained using polar solvents, demonstrating the efficacy of solvents such as water and ethanol for extracting bioactive compounds. According to Mustafa and Turner [33], binary systems may favor the recovery of bioactive compounds because they promote the solute release while increase their solubility, thus, enhancing the extraction yield. This phenomenon was also observed by Cruz et al. [8] during the extraction of bioactive compounds from *P. aculeate* leaves using different solvent mixtures.

The results show that innovative e technologies, such as UAE, are feasible for obtaining bioactive compounds with antioxidant properties from vegetal matrices [34]. In addition, antioxidants obtained from *P. aculeate* leaves are natural and can be utilized as an alternative to synthetic antioxidants. It is also important to mention that previous studies regarding the evaluation of *Pereskia* spp. antioxidant activity presented in the literature did not employ a set of different antioxidant methods, as used in this research, emphasizing the importance of this work since antioxidant compounds can act by distinct mechanisms.

### 3.2. Identification and Quantification of Phenolic Compounds

The phenolic compounds identified and quantified by liquid chromatography corresponded to phenolic acids (chlorogenic acid, gallic acid, caffeic acid, and ellagic acid) and flavonoid (quercetin) groups. Table 2 shows the obtained results.

Figure 1 shows the chromatograms of identified phenolic compounds extracted from the lyophilized material with ethanol solution (50%) and UAE for 20 min. This treatment presented the highest concentrations of chlorogenic acid and the lowest antibacterial activity result value.

The results showed that phenolic acids are found at high concentrations in the *P. aculeate* leaves extracts. The chlorogenic acid, in particular, is the main compound identified in all extracts. High levels of chlorogenic acid were also verified in *Pereskia aculeate* by Agostini-Costa [35] for several genus of *Cactoideae* and Pereskioideae subfamilies.

On the other hand, extracts EOR30 and EOR40 presented the highest contents of gallic and ellagic acids. In addition, caffeic acid was detected only in these treatments. Therefore, the longer exposure time to ultrasound positively contributed to the gallic, caffeic, and ellagic acids extraction efficiency.

Compared to other compounds, the low content of quercetin found in the extracts may be explained by its high instability [36]. This result was also verified by publications in the literature where *P. aculeate* and *Pereskia bleo* (Kunth) leaves extracts were evaluated [19,37]. Finally, catechin was the only compound not detected in any of the *P. aculeate* leaves extracts assessed in this study. Nevertheless, it is important to mention that previous publications described the identification and quantification of this component (9.18 mg.g^−1^) for *P. bleo* ethanolic extract [37]. The absence of catechin in the extracts evaluated here may be explained by factors such as the geographical location of plant cultivation, species, harvest condition, leaves drying, and extraction methods. Moreover, the type of solvent used in the extraction process may directly affect the extraction efficiency because of the solvent polarity and sample profile [38].

### 3.3. Antibacterial Activity Evaluation

Table 3 shows the Minimal Inhibitory Concentration (MIC) of *P. aculeate* leaves extracts against *S. aureus*, *E. coli*, and MRSA bacterial strains. Extracts were evaluated in six different dilutions (50 to 1.56% *v/v*), i.e., from 50 to 1.56 mg.mL^−1^.

As one may notice from Table 3, the extracts presented divergent behaviors for bacterial growth inhibition against Gram-positive bacteria, *S. aureus*, and MRSA. On the other hand, none of them showed antibacterial activity against *E. coli*. Although all the extracts were elaborated using the same raw material and solvent, differences in inhibitory activity between *S. aureus* and MRSA might be related to the resistant phenotype of MRSA and the extraction method used. This result may be associated with the influence of different ultrasound-assisted extraction times as they appear to determine the type and content of *P. aculeate* leaves extracts compounds [39].

All extracts inhibited the *S. aureus* growth, particularly the EO20 treatment that inhibited the growth even at the highest dilution assessed (1.56 mg.mL^−1^). It is important to observe that EO20 treatment produced extracts with higher concentrations of chlorogenic acid (524.54 ± 0.00 μg.g^−1^), in comparison to E30 (459.38 ± 1.00 μg.g^−1^), E40 (364.48 ± 0.03 μg.g^−1^), and E10 (350.93 ± 0.01 μg.g^−1^). Promising results regarding the chlorogenic acid anti-bacterial activity were also obtained in the literature against *S. aureus* and Streptococcus mutans [40,41]. These results suggest that chlorogenic acid had an essential role in anti-bacterial activity against *S. aureus* (E20 showed lower MIC values compared to E10). Furthermore, other bioactive compounds such as caffeic acid (not identified for E10 and E20 treatments) and gallic acid could also benefit MIC values.

The mechanisms that confer MRSA resistance to penicillin, methicillin, and oxacillin (structural alterations of the penicillin-binding proteins—PBP) possibly reduced the compounds’ interaction affinity with the anti-bacterial activity present in the extracts, as observed by MIC values (12.5 to 50 mg.mL^−1^). Gram-negative bacteria, on the other hand, are known because of their high resistance to natural and conventional antibiotics (outer membranes), decreasing the penetration of antibacterial agents [42]. These results are also corroborated by publications in the literature using different types of plant extracts, including *P. aculeate* leaves, where better results for extracts of anti-bacterial activity were registered against Gram-positive bacteria, in comparison with those obtained against Gram-negative ones such as *E. coli* [7,19,43,44,45,46].

Souza et al. [7] also verified the antibacterial activity of *P. aculeate* leaves extracts using disk-diffusion assays. In this publication, the authors found that petroleum ether extract exhibited potent antibacterial activity against *E. coli*, while the chloroformic extract showed inhibitory activity against *Bacillus cereus* and *S. aureus.* The inhibition of *E. coli* growth presented by *P. aculeate* leaves petroleum ether extracts might be related to this solvent’s higher capacity of extracting low molecular weight phenolic compounds with antibacterial activity. However, it is essential to mention that although presenting better performance, those two solvents are toxic and environmental pollutants [47].

Recent studies approaching MIC analysis of *P. aculeate* leaves are scarce. Thus, this study shows that *P. aculeate* leaves extracts present anti-bacterial activity against Gram-positive bacteria. However, the inhibition of microbial growth may be linked to different factors such as the microorganism characteristics, the strains used, extraction conditions, and extracts composition.

### 3.4. Fourier-Transform Infrared (FTIR) Spectroscopy

The FTIR spectroscopy is a diagnostic analysis that provides evidence regarding the presence of functional groups in substances’ chemical structures. FTIR spectroscopy has been widely used for fast multicomponent analyses such as compound identification and structural determination. Thus, the infrared spectrum was obtained for functional groups verification in the lyophilized *P. aculeate* leaves (Figure 2). Similarly, Amaral et al. [48] used this technique to evaluate activated carbon-based structures on *P. aculeate* residue.

As one may notice in Figure 2, N-H bonds (amino group) can be observed at 1646 cm^−1^. The same pattern was observed by Reinas et al. [49] in chitosan infrared spectrograms. Bands at 1440 and 1320 cm^−1^ can be related to CH_3_, CH_2_, flavonoids, and aromatic groups, with C-H vibration and the enlargement of aromatic compounds vibration [50]. The band at 2916 cm^−1^ was characterized as C-H stretch vibrations in cellulose and hemicellulose [51,52]. The bands at 1000 cm^−1^ and 1050 cm^−1^ indicate the gallic acid structural conformation [53], as identified and quantified by chromatography in this study.

The peak observed within the band of 3267 cm^−1^ is characterized by the OH- bond. A similar result has been found in the literature [54], where lyophilized *P. aculeate* fruit mucilage infrared spectrum was evaluated. The presence of OH- bond conferred affinity for polymers in *P. aculeate* leaves mucilage with water molecules, a typical property of hydrophilic compounds [55]. The peak at 1234 cm^−1^ shows lignin’s C-O stretch bond [56].

The peak at the 890 cm^−1^ wave number was identified as characteristic of polysaccharide structures that originated from β-glycosidic bonds between the sugar units. Furthermore, 518 and 942 cm^−1^ bands can be associated with pectins and lignins [57,58]. Finally, the presence of functional carboxyl groups and proteins in the region the between 1000 and 1500 cm^−1^ bands [54,58,59] in *P. aculeate* leaves is evidence of the favoring interaction with water, contributing to crucial characteristics for colloid obtaining, such as gel formation.

### 3.5. Scanning Electron Microscopy (SEM) Analysis

Figure 3 shows SEM images of lyophilized *P. aculeate* leaf’s morphological structure (FOPN). One may notice that the leaves presented an irregular and porous surface, giving the material a broader contact area and contributing to water retention in the product [60,61]. This characteristic is particularly desirable for food production and emulsion formation, as observed by Sato et al. [62] and Conceição et al. [59]. As reported by the authors, this material feature is specifically beneficial for pasta production and obtaining emulsions from *P. aculeate* leaves. The FOPN microscopic analysis showed the presence of paracytic and anomocytic stomata (represented by arrows in Figure 3A,B). The publication by Squena et al. [63] which performed a morpho-anatomical analysis of *P. aculeate* aerial vegetative parts, also showed these structures.

Strong attraction and adhesion of minor and significant particles also indicate an excellent ordination obtained by the lyophilization process. This structure is characteristic of hygroscopic materials such as chitosan and guar gum [64,65]. The FOPN microstructures, such as porous and irregular surfaces, revealed characteristics similar to those of Monteiro et al. [66] in lyophilized *P. aculeate* leaves. The obtained microstructure is typical of ice sublimation formed in intracellular spaces. These microstructures are responsible for maintaining the leaf cellular structure, leading to particles with irregular surfaces formed by large pores [66].

As presented in Figure 3, the lyophilized *P. aculeate* leaves presented a rough surface. It was observed that more prolonged exposure to ultrasound-assisted extraction mediated the cavitation process in the vegetal matrix [30], releasing more significant quantities of phenolic acids. Considering these results, one may infer that the lyophilization process promoted the conservation of morphological structures of *P. aculeate* leaves, granting more versatility and stability for their components, mainly used for formulation purposes as an additive, for instance, in pharmaceutical and food industries. 

### 3.6. Chemical Profile by Paper Spray Mass Spectrometry (PS-MS)

The PS-MS analysis was used to obtain the chemical fingerprint of complex matrices present at *P. aculeate*. This technique provides a super quick and low-cost methodology for assessing the overall quality of the material without generating chemical residues. Thus, the extract obtained by ethanol solution (50%) and ultrasound-assisted extraction for 20 min was chosen for chemical profile evaluation. The chosen extract, presenting the best results for antibacterial activity, was analyzed by PS-MS in positive and negative ionization modes, as shown in Figure 4.

The characterization in negative and positive ionization modes revealed 53 chemical compounds present, including organic, fatty, and phenolic acids. Additionally, sugars, flavonoids, terpenes, phytosterols, and other components were identified. Comparably, Garcia et al. [19] identified minor compounds in the *P. aculeate* leaves hydroethanolic extract (10 substances). The authors reported two phenolic acids (caffeic acid derivatives) and eight flavonoids (quercetin, kaempferol, and isorhamnetin glycoside derivatives) were identified. Table 4 shows the 33 substances identified for EO20 treatment using the negative ionization mode.

As one may notice from Table 4, the group of flavonoids was the major compound group quantified in the negative ionization mode. Similar results were reported in the publication by Silva et al. [47], where the *Eriobotrya japonica* Lindl chemical profile was characterized by PS-MS. This result is in accordance with the class of secondary plant metabolites represented by flavonoids. These substances have important health benefits, such as antioxidant activity [79].

According to Li et al. [76], the quercetin-3-*O*-glucoside (*m/z* 463 [M-162-H]-) and rutin (*m/z* 609 [M-308-H]-) MS/MS fragmentation generated the *m/z* 301 ions, corresponding to quercetin without the glucoside and rutinoside units, respectively. In the present study, the main flavonoids found in *P. aculeate* leaves were detected by PS-MS, namely quercetin, isorhamnetin, and kaempferol [35]. However, kaempferol-xylose is classified as a glycoside, a common characteristic of *P. aculeate* in different parts, which also presents various aglycones.

Although HPLC did not detect caffeic acid and quercetin, the PS-MS analysis identified those compounds. This result highlights the technique’s ability to evaluate compound fingerprints, covering an extensive molecular mass range, and to identify compounds that often are not detected by other methods. Souza et al. [78] also found caffeic acid (*m/z* 179) and ferulic acid (*m/z* 193) in *P. aculeate in natura* leaves. 5-feruloylquinic acid was characterized as an ester of ferulic acid and quinic acid.

To the best of our knowledge, this is the first study that identified substances from each chemical class of organic acids, fatty acids, phenolic acids, sugars, flavonoids, and terpenes in lyophilized *P. aculeate* leaves extract obtained by an environmentally friendly process technology.

Using the positive ionization mode, 20 distinct compounds were identified, including flavonoids, phytosterols, terpenes, and sugar chemical classes. Table 5 shows the identified compounds using positive ionization mode for EO20 samples.

Again, the primary group identified in the samples was the group of flavonoids. Most of these flavonoids were found to be conjugated derivatives of glycosides. As expected, the conjugated form of these compounds is a typical characteristic of the leaves because it provides better plant protection [81].

Pinto et al. [11] used the positive mode of HPLC-MS/MS to detect only seven different compounds (tryptamine, abrine, mescaline, hordenine, petunidin, di-tert-butylphenol isomers, and quercetin) in *P. aculeate* leaves extract samples. The authors also evaluated the antinociceptive activity of extracts obtained by hydromethanolical solutions. The discrepancy found in the results, using the same positive ionization mode, indicated that the use of ultrasound-assisted extraction enhanced the bioactive compounds obtained, in comparison to traditional extraction processes [34,87].

## 4. Conclusions

The present work presented a set of methods for evaluating antioxidant compounds that have not yet been verified in studies of Pereskia aculeate Miller leaves. We were able to verify that the content of phenolic compounds and antioxidant activity are influenced by different extraction times in an ultrasonic bath. The ora-pro-nobis leaf extracts showed antibacterial activity against *S. aureus* and MRSA strains, also demonstrating a potential use in formulations with against these Gram-positive bacteria. EO20 extract, for example, inhibited *S. aureus* up to the last dilution evaluated (1.56 mg.mL^−1^) and showed the highest content of chlorogenic acid, a substance with described antibacterial activity.

A total of 53 compounds were found in *P. aculeate* leaves extracts applying the PS-MS technique, proving to be a source of organic, fatty, and phenolic acids, flavonoids, terpenes, phytosterols, sugars among other substances with important antioxidant and antimicrobial activities. Employing FTIR and SEM, it was possible to verify the presence of functional groups that help the formation of colloids, which are desirable in pharmaceutical and food formulations in industries, and that lyophilization was able to promote greater conservation of the morphological structures of ora-pro-nobis leaves. As a perspective, extracts from the leaves of P. aculeate Miller can be used in nutraceuticals and functional foods, due to the high levels of polyphenols, including their antioxidant and antibacterial activities.

## Figures and Tables

**Figure 1 metabolites-13-00691-f001:**
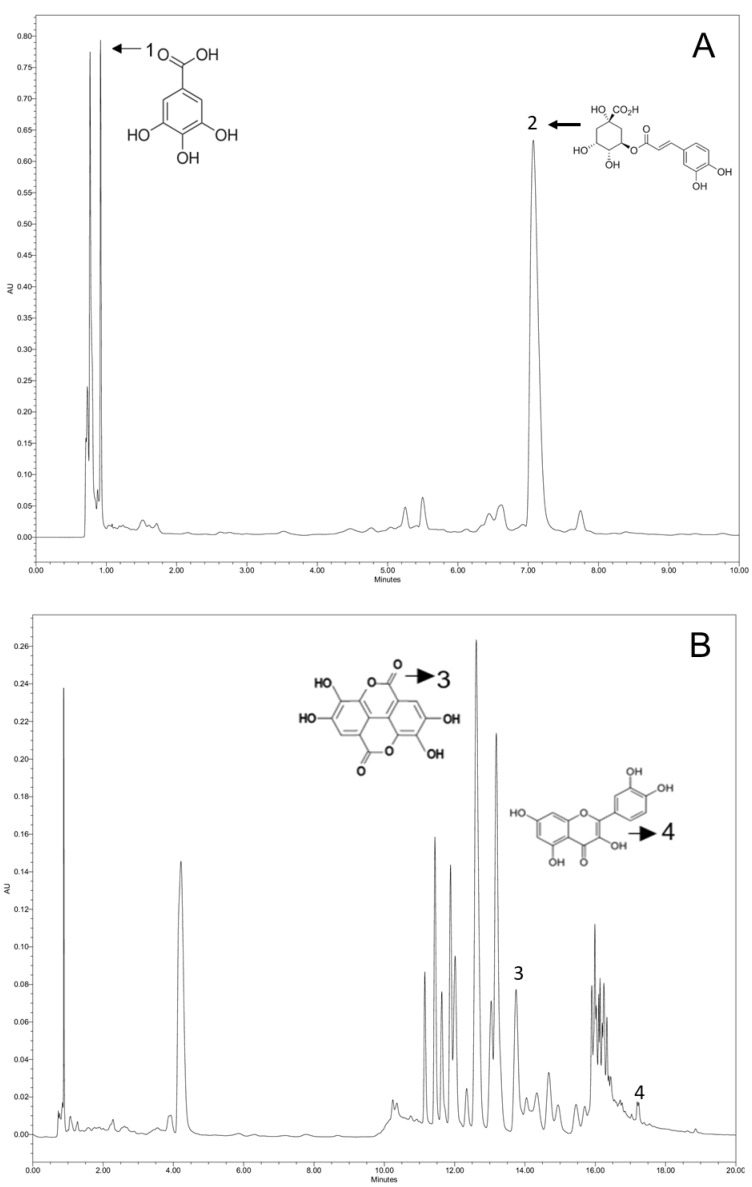
Identified phenolic compounds chromatograms in the lyophilized *P. aculeate* leaves extracted with ethanol (50%) and ultrasound for 20 min (EO20). (**A**): spectrum of standard pattern diluted in water; 1: gallic acid; 2: chlorogenic acid; (**B**): spectrum of standard pattern diluted in methanol; 3: ellagic acid; 4: quercetin.

**Figure 2 metabolites-13-00691-f002:**
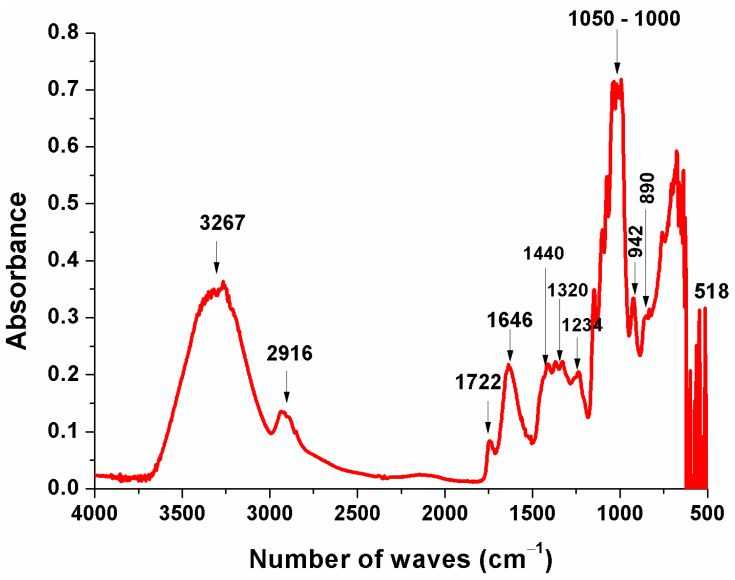
Lyophilized *P. aculeate* leaves Infrared spectrum with Fourier transform.

**Figure 3 metabolites-13-00691-f003:**
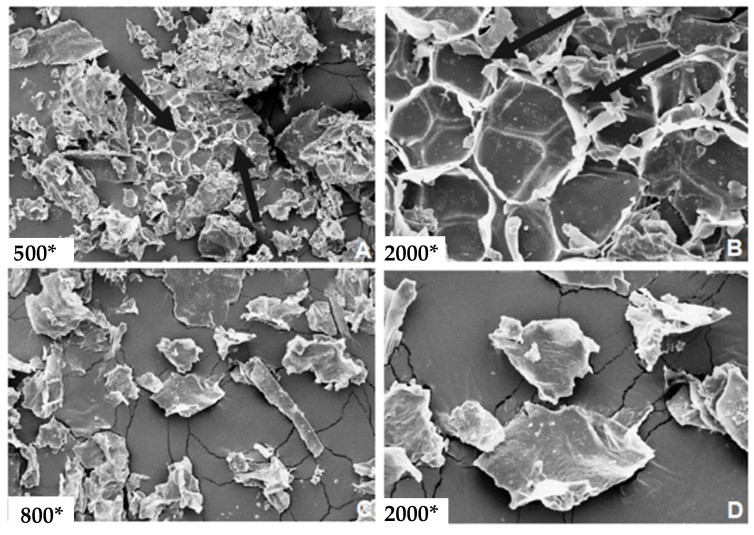
Lyophilized *P. aculeate* leaves SEM images. * sign for changed multiplication. The legend is unique in that it is the same figure with different sources of extension, as evidenced in each subfigure (**A**–**D**).

**Figure 4 metabolites-13-00691-f004:**
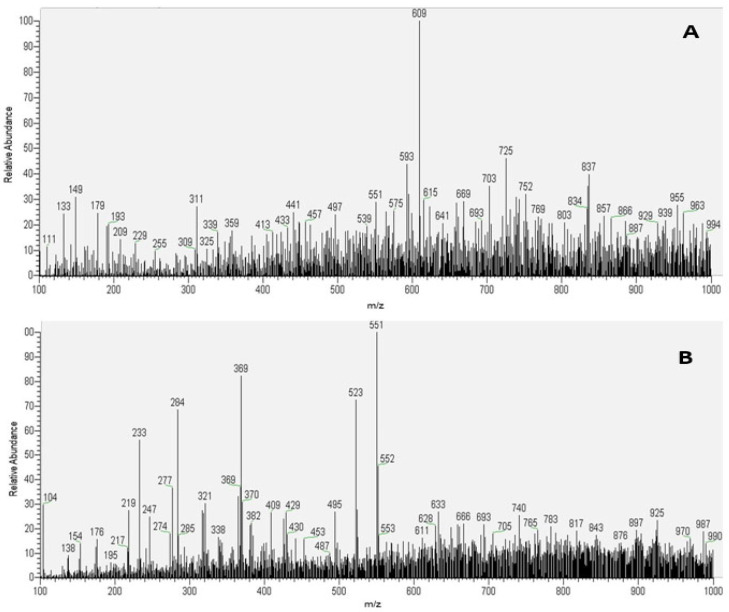
Paper spray mass spectra representation. (**A**) PS-MS spectrogram in negative ionization mode and (**B**) PS-MS spectrogram in positive ionization mode of the lyophilized *P. aculeate* leaves extract extracted with ethanol (50%) by ultrasound for 20 min (EO20).

**Table 1 metabolites-13-00691-t001:** Total phenolic compounds (TPC) and antioxidant activity (ATT) of *P. aculeate* leaves hydroalcoholic extracts.

Method	EO10 *	EO20 *	EO30 *	EO40 *
**TPC** mg EAG.g^−1^ of extract	2.35 ± 0.03 ^ab^	2.14 ± 0.04 ^bc^	2.07 ± 0.07 ^c^	2.60 ± 0.06 ^a^
**AAT by ABTS** *^+^ (μM de TE.g^−1^ of extract)	10.24 ± 0.40 ^a^	9.00 ± 0.08 ^b^	6.38 ± 0.17 ^c^	7.14 ± 0.17 ^c^
**AAT by FRAP** (μM ferrous sulp.g^−1^ of extract)	30.00 ± 0.27 ^ab^	24.34 ± 0.87 ^c^	32.13 ± 0.60 ^a^	27.37 ± 0.83 ^bc^
**AAT by DPPH** (μM of TE.g^−1^ of extract)	24.48 ± 0.15 ^b^	28.22 ± 0.99 ^b^	70.20 ± 1.05 ^a^	61.20 ± 1.12 ^a^

* Means value ± standard deviation; n = 3. Equal letters on the same line do not differ significantly from each other at 5%, by Tukey’s test (*p* < 0.05). Aqueous extracts, named EO10, EO20, EO30 and EO40 corresponding to 10, 20, 30 and 40 min ultrasound treatment, respectively; AAT: total antioxidant activity; TE: Trolox equivalent; EAG: equivalents acid gallic.

**Table 2 metabolites-13-00691-t002:** Phenolics profile (μg.g^−1^ of extract) in *P. aculeate* leaves hydroalcoholic extracts.

Compound	EO10 *	EO20 *	EO30 *	EO40 *
**Chlorogenic acid**	350.93 ± 0.01 ^d^	524.54 ± 0.00 ^a^	459.38 ±1.00 ^b^	364.48 ±0.03 ^c^
**Gallic acid**	0.49 ± 0.17 ^d^	0.31± 0.09 ^c^	12.01 ± 0.20 ^b^	14.66 ± 0.11 ^a^
**Caffeic acid**	ND	ND	29.03 ± 0.04 ^b^	31.23 ± 0.02 ^a^
**Ellagic acid**	1.00 ± 0.25 ^d^	1.35 ± 0.02 ^c^	6.32 ± 0.00 ^b^	10.01 ± 0.01 ^a^
**Quercetin**	0.05 ± 0.03 ^d^	0.10 ± 0.05 ^a^	0.08 ±0.01 ^c^	0.09 ± 0.08 ^b^
**Catechin**	ND	ND	ND	ND

* Mean value ± standard deviation; n = 3. Equal letters on the same line do not differ significantly from each other at 5%, by Tukey’s test (*p* < 0.05). Aqueous extracts, named EO10, EO20, EO30 and EO40 corresponding to 10, 20, 30 and 40 min ultrasound treatment, respectively; ND: not detected.

**Table 3 metabolites-13-00691-t003:** Minimal inhibitory concentration (mg.mL^−1^) of the *P. aculeate* leaves hydroalcoholic extracts against *Staphylococcus aureus*, *Escherichia coli* and methicillin resistant *Staphylococcus aureus* (MRSA).

*Staphylococcus aureus* (ATCC 29213)				*MRSA* (ATCC 43300)				*Escherichia coli* (ATCC 35218)			
EO10	EO20	EO30	EO40	EO10	EO20	EO30	EO40	EO10	EO20	EO30	EO40
6.25	1.56	3.13	6.25	25	12.5	50	NA	NA	NA	NA	NA

NA: Not active. EO10, EO20, EO30 and EO40 correspond to 10, 20, 30 and 40 min ultrasound treatment, respectively.

**Table 4 metabolites-13-00691-t004:** Identified ions in lyophilized *P. aculeate* leaves extract obtained from extraction with ethanol (50%) by ultrasound for 20 min (EO20) by PS-MS in negative ionization mode.

Compound	*m/z*	MS/MS	Reference
**Organic acids**			
Fumaric acid	115	71	(Al Kadhi et al. [67])
Mallic acid	133	115	(Abu-Reidah et al. [68]; Silva et al. [27])
Cumaric acid	163	119	(Abu-Reidah et al. [68]; Sun et al. [69])
Ferulic acid	193	149	(Wang et al. [30])
Fertaric acid	325	163,193	(Aaby et al. [70]; Abu-Reidah et al. [68])
**Fatty acids**			
Stearic acid	283	237	(Wang et al. [30])
Ricinoleic acid	297	183	(Wang et al. [30])
Eicosanoic acid	311	293	(Wang et al. [30])
Trihydroxy-octadecadienoic acid	327	291,309	(Kang et al. [71])
**Phenolic acids**			
Caffeic acid	179	135	(Kang et al. [71])
Quinic acid	191	127,173	(Abu-Reidah et al. [68]; Chen et al. [72])
5-feruloylquinic acid	367	175	(Zhang et al. [73])
Apigenin-6-C-glucoside	431	431	(Kang et al. [71])
Eriodictiol 6,8 di-C-glucoside flavonoid	611	491,593	(Simirgiotis et al. [74])
**Sugars**			
Hexose	215	179	(Guo et al. [75]; Silva et al. [27])
Saccharide	371	113,121,231,249	(Kang et al. [71])
**Flavonoids**			
Kaempferol–xylose	417	152,285	(Chen et al. [72])
Quercetin-3-rhamnoside	447	447,300	(Zhang et al. [73])
Verbonol	453	435	(Abu-Reidah et al. [68])
Quercetin-3-*O*-glucoside	463	300,301,343	(Li et al. [76]; Zhang et al. [77])
Taxifolin hexoside	465	303,447	(Kang et al. [71])
Dimethyl Ellagic acid hexoside	491	454	(Silva et al. [27])
Caffeoyl derivative hexose	499	337	(Kang et al. [71])
Kaempferol-3-*O*-rutinoside	593	285,447	(Silva et al. [27]; Wang et al. [30])
Rutin	609	255,271,301,463	(Chen et al. [72]; Silva et al. [27]; Wang et al. [30])
Naringin	579	271,459	(Sun et al. [69])
Isorhamnetin-3-*O*-rutinoside	623	315	(Souza et al. [78])
Vicenin II derivative	629	593	(Simirgiotis et al. [74])
**Terpenes**			
Sorhamnetin-7-*O*-glucopyranoside	477	243,343	(Wang et al. [30])
Metil corosolate	485	423,467	(Chen et al. [72])
**Others**			
Spinochrome A	265	235	(Abu-Reidah et al. [56])
Plastoquinone 3	339	135,163,203	(Souza et al. [78])
Vaccihein A	377	289,347,361	(Souza et al. [78])

**Table 5 metabolites-13-00691-t005:** Identified ions in lyophilized *P. aculeate* leaves extract obtained from extraction with ethanol (50%) by ultrasound for 20 min (EO20) by PS-MS in positive ionization mode.

Compounds	*m/z*	MS/MS	Reference
**Flavonoids**			
3-*O*-methylquercetin	317	121,193,245,274	(Gobbo-Neto and Lopes [80]; Silva et al. [81])
6,8-di-C-β-glicopiranosil cristine	579	495,507,543,561	(Gobbo-Neto and Lopes [80])
Lucenine-2 (6,8-di-C-β-glucopiranosilluteoline	611	527,593	(Gobbo-Neto and Lopes [80])
Rutin	633	331, 487,615	(Jia et al. [82])
Tricin di-*O*,*O*-hexosídeo	655	493	(Silva et al. [81])
Sagittatoside A	677	531	(Ren and Long [83])
3-Hidroxicariine-*O*-glucose-rhamnose	693	547	(Ren and Long [83])
lolidide-β-*D*-glucopyranoside	711	549,651,693	(Jia et al. [82])
Chrysoeriol *O*,*O*-malonyl hexoside	797	711	(Cavaliere et al. [84])
**Phytosterols**			
Sitosterol	397	247	(Wang et al. [30])
**Terpenes**			
Deacetylforskolin	369	235,431	(Abu-Reidah et al. [68]; Zhang et al. [85])
Vomifoliol β-*D*-	409	353,391,394	(Jia et al. [71])
Dihidroisovaltrato	425	365	(Abu-Reidah et al. [68])
Ononin	431	269,431	(Ren and Long [83])
**Sugars**			
Sacarose	365	185	(Guo et al. [75])
Morroniside	429	267	(Guo et al. [75]; Xiong et al. [86])
**Phenolic Acids**			
Licanic acid	293	257,275	(Wang et al. [30])
**Fatty Acids**			
Magnoflorine	342	282,342	(Ren and Long [83])
**Others**			
L arginine	175	129	(Silva et al. [27])
Diallyl phthalate	247	187	(Ren and Long [83])

## Data Availability

Not applicable.

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
