# Peer review of "Elaboration and Characterization of *Pereskia aculeate* Miller Extracts Obtained from Multiple Ultrasound-Assisted Extraction Conditions"

_metabolites, 2023, doi:10.3390/metabo13060691_

Round 1

Reviewer 1 Report

Please, see attachment.

Author Response

  • Reviewer 1: 1. A 4-lines-long title seems excessive, the title should be concise, specific and relevant.
    We appreciate the suggestion. As recommended by the reviewer, we rewrote the manuscript title adopting the given guidelines.
  • Reviewer 1: 2. English language must be revised and improved: sentences are most times too long, punctuation is often incongruous, sentence construction is sometimes inappropriate making it difficult to interpret.
    Thank you again for the observation. As suggested, the whole manuscript has been thoroughly reviewed by a native English speaker. We hope this rewritten version meets your approval.
  • Reviewer 1: 3. Lines 105-108 are redundant, and the same is repeated in Tables headings. Something like "...resulting in aqueous extracts, named EO10, EO20, EO30 and EO40 corresponding to 10, 20, 30 and 40 minutes ultrasound treatment, respectively" will suffice.
    The authors thank the reviewer’s suggestion. As recommended, we replaced these sentences and made them less redundant.
  • Reviewer 1: 4. Section 2: Methods should be described in deeper detail, in order to assure reproducibility in other laboratories and by other operators.
    We appreciate the reviewer’s suggestion. However, we included in the manuscript references where the reader may find a thorough description of methods’ procedures. It is important to mention that, for the sake of manuscript’s concision, it was necessary to avoid extensive descriptions of the methods. Nevertheless, we have reviewed the methods section, adding more detailed descriptions and references.
  • Reviewer 1: Section 3.1: Readers could benefit of a brief discussion of the chosen protocols.
    We appreciate the reviewer’s observation. As suggested we included a brief discussion rearding the rational used to choose the protocols for evaluating the total antioxidant activity, as follows
    It is essential to mention that the protocols employed in the antioxidant activity assessment were chosen because of their complementary features. Hence, it was possible to identify a multitude of bioactive compounds performing this activity in the samples.”
  • Reviewer 1: Section 3.2: Please describe how the compounds were identified (by comparison with standards?); similarly, describe how the concentration of those compounds was determined (internal standard - single point or a calibration curve? External standard - single point or a calibration curve)? Standard addition method - different methods ?)
    As suggested by the reviewer, the mentioned information was included in section 2.6, as follows:
    High-performance liquid chromatography (HPLC) analyses were conducted accord-ing to the method described by Eça et al. [16]. The four extracts were filtered in a 0,22 μm nylon filter syringe and injected in HPLC equipment (Acquity UPLC® Class, Milford, Wa-ters, Massachusetts, USA), equipped with a UV detector by diode array, quarternary bomb, degasser, autosampler, and column (Acquity UPLC ® BEH C18 - 2,1 x 100 mm; 1,7 μm, Waters, Milford, Massachusetts, EUA). The flow was kept constant at 0,3 mL.min-1, with two mobile phases (A = acetonitrile and B = water: formic acid, 99,75: 0,25).
    Calibration curves used to identify the compounds were prepared with the following external standards: gallic acid, catechin, and chlorogenic acid (diluted in water for com-plete dissolution); caffeic acid, ellagic acid, and quercetin (diluted in methanol for com-plete dissolution). While the water diluted standards were run for 10 min in an isocratic elution mode of 5: 95 v/v (A: B), a linear gradient was applied to the other standards: 0 min, 8% (A); 8 min, 15% (A); 14 min, 25% (A); 20 min, 8% (A). The UV spectra were ob-tained between 200 and 400 nm. Chromatograms were processed at 253, 271, 320, and 372 nm. The calibration curves were built within the following ranges: gallic acid and querce-tin (21.2 to 212 μg.g-1), catechin (24.4 to 244 μg.g-1), chlorogenic acid (18.4 to 184 μg.g-1), and ellagic acid and caffeic acid (20 to 200 μg.g-1). All data acquired from these analyses were processed in the Empower® software, and the results were expressed in μg.g-¹ of ex-tract.”
  • Reviewer 1: Section 3.4: Why did the authors choose IR? Why not NMR or other techniques? Please, briefly discuss the advantages of the chosen method
    As suggested by the reviewer a brief discussion was added to the mauscript, explaining the advantages of using FTIR, as follows:
    The Fourier transform infrared spectroscopy (FTIR) is a diagnostic analysis that provides evidence regarding the presence of functional groups in substances' chemical structures. FTIR spectroscopy has been widely used for fast multicomponent analyses such as compound identification and structural determination.”
  • Reviewer 1: Section 3.5: The authors should clarify the aim and purpose of SEM analysis and how it is relevant to this study. Is the study focused on the leaves extracts?
    The authors appreciate the reviewer’s observation. Indeed, the SEM analysis was used to verify the morfological structure of P. aculeata leaves. This evaluation is of paramount importance for this study because it allows the assessment of the influence of ultrasound-assisted extraction technique on the releasing process of phenolic compounds. This information was discussed in the manuscript as follows:
    As presented in Figure 3, the lyophilized P. aculeate leaves presented a rough surface. It was observed that more prolonged exposure to ultrasound-assisted extraction mediated the cavitation process in the vegetal matrix [21], releasing more significant quantities of phenolic acids. Considering these results, one may infer that the lyophilization process promoted the conservation of morphological
    structures of P. aculeate leaves, granting more versatility and stability for their components, mainly used for formulation purposes as an additive, for instance, in pharmaceutical and food industries.”
  • Reviewer 1: Following section: The authors should discuss why they have chosen PS-MS over other mass spectrometry techniques, such as ESI-MS. What are the advantages and superior properties of the chosen technique?
    Thank you again for your valuable observations. As suggested by the reviewer, we included in the manuscript a short discussion concerning the advantages of using PS-MS, as follows:
    The PS-MS analysis was used to obtain the chemical fingerprint of complex matrices present at P. aculeate. This technique provides a super quick and low-cost methodology for assessing the overall quality of the material without generating chemical residues.”
  • Reviewer 1: 11. Table formatting should be uniform; smaller fonts may be used, similar to the main text in size and line spacing.
    As recommended by the reviewer, we made the suggested adjusts in the mentioned table.
  • Reviewer 1: 12. Section 4: it appears somehow presumptuous to state in your own conclusions that this study stands out. Please stick to facts, using scientific and neutral language.
    We appreciate the reviewer’s comment. Indeed, this revised manuscript version presents a new conclusion text, attending the reviewer’s directions.

Reviewer 2 Report

Title is so long, and also the biological activity is only for antibacterial activity.. the title should indicate the most relevant or major in the study.

Introduction:

lines 56 to 59 are confusing in order to relate to the objective of this study.

also so many paragraph are interrupted by space, the text and idea should be continuos in some paragrpah on the text for a better understanding.

Materials and methods

Plant sampling: how does the plant sampling was realized? how many plants were taken? details about the physiological stage of the leaves taken for the study should be indicated.

Antioxidant activity units are GAE: gallic acid equivalents

Indicate in methods the internal and standard control use for the identification of the metabolites, also the LOD or limit values used.

Results and discussion

Table 1. indicate in the tittle the AAT abbreviation

Table 1. check the deviation standar on the TPC, the values are higher than the measure, so this indicate a high error. with this values the measure are not significative.

Table 1. units for ABTS and DPPH are GAE, units on FRAP should be indicated in a better way

-The results of the antioxidant activity are not adequately described, as explained by the observed values, what was the need to use three methods and if it is necessary to indicate their mechanisms based on what they observed in the composition, it could be used later.

-lines 212-216 are not clear

-line 224 indicates correlation in table 1 but that information does not appear as such in the table, it can only be seen in the table that there is an increase in some of the AAT evaluations with respect to treatment but there are no correlation values or any evaluation that indicates that this is significant-how do you explain that the values are similar at lower times with respect to high times in some evaluations? why not included a control without treatment for check if the treatment is neccesary, only specualitions regarding this is indicate on the text from the SEM analysis.

-Table 2. it is not necessary indicate catechin because is not detected 

- In some points of results and discussion flavonoids are indicated,  did you consider that the total content of flavonoids could have been evaluated to contrast with the content of total phenolics?

lines 314-317 indicate a relationship between the levels of cholorogenic and antimicrobial activity but a better description is required

figures 1 and 2 should be improved for a better appreciation

conclusion need to be improved regarding to results and perspectives 

Author Response

  • Reviewer 2: Title is so long, and also the biological activity is only for antibacterial activity.. the title should indicate the most relevant or major in the study.
    We appreciate the suggestion. As recommended by the reviewers, this revised manuscript version counts with a more concise title.
  • Reviewer 2: Lines 56 to 59 are confusing in order to relate to the objective of this study: The authors thank the reviewer’s observation. As suggested, the mentioned lines were revised. In this revised version, they highlight the importance of obtaining extracts to be used in the food/pharmaceutical industries and the promising potential of the extraction technique used in this study. We hope this new version meets the reviewer approval.
  • Reviewer 2: Also so many paragraph are interrupted by space, the text and idea should be continuos in some paragrpah on the text for a better understanding: As suggested by the reviewer, the text structure was reformulated. In its revised version, the manuscript displays the main discussions and ideas in a rational order to better present the data to the reader. Moreover, we thoroughly reviewed the manuscript’s English language.
  • Reviewer 2: Plant sampling: how does the plant sampling was realized? how many plants were taken? details about the physiological stage of the leaves taken for the study should be indicated.
    Thank you for this observation. As recommended, the information was added to the manuscript, as follows:
    The P. aculeate leaves were selected according to their visual appearance (typical green color and integrity). After selection, the leaves were sanitized with flowing water and dried with paper towels. Thus, the material was submitted to freezing in an Ultra Low-Temperature Freezer at -55 °C for 24 h.”
  • Reviewer 2: Antioxidant activity units are GAE: gallic acid equivalents: As observed by the reviewer, this information was corrected in the manuscript, as follows:
    Following the Folin-Ciocalteu method [13], the total phenolic compounds content was determined, with results expressed in equivalents of gallic acid (mg EAG.g-1 of the sample).”
  • Reviewer 2: Indicate in methods the internal and standard control use for the identification of the metabolites, also the LOD or limit values used.
    The authors appreciate the reviewer’s recommendation. The mentioned information was included in the manuscript’s section 2.6 as follows:
    Calibration curves used to identify the compounds were prepared with the following external standards: gallic acid, catechin, and chlorogenic acid (diluted in water for com-plete dissolution); caffeic acid, ellagic acid, and quercetin (diluted in methanol for com-plete dissolution). While the water diluted standards were run for 10 min in an isocratic elution mode of 5: 95 v/v (A: B), a linear gradient was applied to the other standards: 0 min, 8% (A); 8 min, 15% (A); 14 min, 25% (A); 20 min, 8% (A). The UV spectra were ob-tained between 200 and 400 nm. Chromatograms were processed at 253, 271, 320, and 372 nm. The calibration curves were built within the following ranges: gallic acid and querce-tin (21.2 to 212 μg.g-1), catechin (24.4 to 244 μg.g-1), chlorogenic acid (18.4 to 184 μg.g-1), and ellagic acid and caffeic acid (20 to 200 μg.g-1). All data acquired from these analyses were processed in the Empower® software, and the results were expressed in μg.g-¹ of ex-tract.
  • Reviewer 2: Table 1. indicate in the tittle the AAT abbreviation: As recommended by the reviewer, this information was included in the table’s caption.
  • Reviewer 2: Table 1. check the deviation standar on the TPC, the values are higher than the measure, so this indicate a high error. with this values the measure are not significative: The reviewer raised a valid point, thank you. The authors revised the statistics of data presented in Table 1.
  • Reviewer 2: Table 1. units for ABTS and DPPH are GAE, units on FRAP should be indicated in a better way
    As suggested, the units’ indications were remade. We included clearer indication in the methods text and in the table.
  • Reviewer 2: The results of the antioxidant activity are not adequately described, as explained by the observed values, what was the need to use three methods and if it is necessary to indicate their mechanisms based on what they observed in the composition, it could be used later.
    Thank you for raising this point. As observed by the reviewer, the rational for choosing the AAT methods was described in the manuscript as follows:
    It is essential to mention that the protocols employed in the antioxidant activity as-sessment were chosen because of their complementary features. Hence, it was possible to identify a multitude of bioactive compounds performing this activity in the samples.”
  • Reviewer 2: lines 212-216 are not clear
    Thank you for this observation. The mentioned sentences were rephrased, and we hope they meet the reviewer’s approval.
  • Reviewer 2: -line 224 indicates correlation in table 1 but that information does not appear as such in the table, it can only be seen in the table that there is an increase in some of the AAT evaluations with respect to treatment but there are no correlation values or any evaluation that indicates that this is significant-how do you explain that the values are similar at lower times with respect to high times in some evaluations? why not included a control without treatment for check if the treatment is neccesary, only specualitions regarding this is indicate on the text from the SEM analysis.
    The authors appreciate and considered the reviewer’s comment. Indeed, we did not consider including a control treatment in the assays. To cover the reviewer’s questions we revised the discussion section including more evaluations and comparisons with literature data.
  • Reviewer 2: -Table 2. it is not necessary indicate catechin because is not detected: As observed by the reviewer, this compound analysis results were included in Table 2, even though it was no detected, to indicate that it was evaluated in the extracts.
  • Reviewer 2: In some points of results and discussion flavonoids are indicated, did you consider that the total content of flavonoids could have been evaluated to contrast with the content of total phenolics? The reviewer raised a valid point. However, the mentioned comparison could not be conducted because of technical and local limitations of the group. To consider this demand, we included a brief discussion about the qualitative analysis performed with PS-MS.
  • Reviewer 2: lines 314-317 indicate a relationship between the levels of cholorogenic and antimicrobial activity but a better description is required
    As suggested by the reviewer, this section was completely revised to better explain the correlations.
  • Reviewer 2: figures 1 and 2 should be improved for a better appreciation: The authors thank the reviewer’s observation. Indeed, we made changes to better present the mentioned figures. However, it is important to mention that images used in these figures were captured directly from the equipment processing software. Hence, it was not possible to significantly increase their resolution.
  • Reviewer 2: Conclusion need to be improved regarding to results and perspectives
    We appreciate the suggestion. Thus, the conclusion section was revised following the reviewer’s directions.

Reviewer 3 Report

The manuscript entitled "Phytochemical profile determination, biological and antioxidant activities and microscopical characterization of the Pereskia aculeata Miller leaves and extracts obtained from different ultrasonic extraction times" has made an attempt to characterize the bioactive compounds of the plant also analyzed their biological properties. Though, the study contains interesting information, there needs extensive modification prior to further consideration in a journal like metabolites. My specific comments.

Abstract

Authors need to indicate the common name and scientific name first and use any abbreviation of the scientific name or as "P. aculeata" thereafter. No need to repeat the common name in the whole manuscript.

Authors need to include the major compounds  and also include quantitative data in the abstract

A background information of the study needs to be included in the initial line of abstract

Introduction

A description on the plants, phytochemicals and their importance in food and medicine may be included as the beginning paragraph

Authors may expand the introduction by describing the importance of the family to which the P. aculeata belongs.

More detailed information is needed on the plant

Why the authors chose the ultrasound based extraction? Literature have indicated that the ultrasound assisted extraction increases the yield of phytomolecules and thereby increase the biological efficacy of the isolate/ extract. Authors need to include a detailed account on the methodology and significance of ultrasound based method of extraction. Authors may refer and include the following articles in the paragraph;

https://doi.org/10.1016/j.foodres.2022.111268

https://doi.org/10.1016/j.ultsonch.2016.06.035

https://doi.org/10.1016/j.envres.2021.111718

https://doi.org/10.1016/j.pmpp.2021.101746

https://doi.org/10.1016/j.ultsonch.2020.105325

https://doi.org/10.3390/molecules27051456

https://doi.org/10.3389/fnut.2022.870923

https://doi.org/10.3390/app122010446

https://doi.org/10.3390/agronomy7030047

https://doi.org/10.1155/2020/3497107

The line "Furthermore, studies show that ora-pro-nóbis extracts and leaves are also rich in carotenoids and phenolic compounds, providing healing, anti-inflammatory and antioxidant activities, as well as analgesic potential and antifungal reaction [3-7]" needs to be expand with individual literature support.

Authors can get more biological activities of the plant such as 

Journal of Ethnopharmacology 194 (2016): 131-136

Food chemistry 294 (2019): 302-308.

Food Science and Technology 39 (2018): 28-34

Journal of Ethnopharmacology 173 (2015): 330-337.

Journal of medicinal food 19.9 (2016): 890-894.

Ultrasonics sonochemistry 50 (2019): 339-353.

Food Chemistry 361 (2021): 130078

Overall, the introduction needs to be re-written completely.

Materials and Methods

Authors need to incorporate the voucher specimen number of the plant and identification details

As I suggested earlier, the common name of the plant is relevant to the country only. So replace the same with scientific name or abbreviation

The title emphasized "Ultrasonic extraction", hence, the details must be given including minute points about the frequency and set of instrument

While mentioning "Gram- positive or Gram- negative" the "G" must be in upper cases

While mentioning instruments, the model, make, city and country details must be included

Results

The table 3 must be formatted to indicate the MIC value only

Why the authors not estimated the zone of inhibition by disc diffusion assay; it needs to be done and result is to be included

The selected strains (3 numbers) seems to be insufficient. Authors include data on one or two more strains

In Figure 2, the markings on peaks are not clear; improve the figure quality. Also the direction of Y-Axis legend may be changed and remove the title inside the figure

Discussion

The discussion contains limited literature support. I feel combining results and discussion makes it difficult to include a detailed analysis of the observed information and comparison with the available literature. Hence, I suggest to separate these two sections as stand alone sections

The formating of reference needs to be cross checked; there are some errors in the reference section (formating and punctuation errors)

Author Response

  • Reviewer 3: Authors need to indicate the common name and scientific name first and use any abbreviation of the scientific name or as "P. aculeata" thereafter. No need to repeat the common name in the whole manuscript: We appreciate the reviewer’s suggestion. As suggested, the abbreviation ‘P. aculeata’ was adopted in the text.
  • Reviewer 3: Authors need to include the major compounds and also include quantitative data in the abstract: By liquid chromatography, chlorogenic acid was the compound detected in greater quantity for all extracts. With paper spray mass spectrometry (PS-MS) was possible to suggest the presence of 53 substances, such as: organic and phenolic acids, flavonoids, and others: We appreciate the suggestion. However, the major compounds were presented throughout the manuscript in a general way, not highlighting any specific one. Nevertheless, we considered the reviewer’s recommendation and rewrote this section.
  • Reviewer 3: A background information of the study needs to be included in the initial line of abstract: Added.
  • Reviewer 3: A description on the plants, phytochemicals and their importance in food and medicine may be included as the beginning paragraph: As suggested, an introductory paragraph was included in the manuscript, as follows:“Plant materials are natural sources of phytochemicals. In the human body, these compounds perform crucial roles such as protection, antioxidant, anti-inflammatory, an-tifungal, and anti-bacterial activities. On top of that, phytochemicals act in the immune system stimulation [1,2].
  • Reviewer 3: Authors may expand the introduction by describing the importance of the family to which the P. aculeata belongs: As suggested, a descriptive paragraph was included in the manuscript, as follows: "Pereskia aculeate Miller, known worldwide as ora-pro-nobis, is a South American un-conventional food plant, known by the acronym UFP (PANC, in Portuguese), belonging to the Cactaceae family and Pereskioideae subfamily. The plant’s leaves, its non-toxic edible portion, are used in traditional medicine and cooking [1,2]. The P. aculeate leaves are a nutritional complement primarily because of their high protein, fiber, calcium, and iron contents. Furthermore, P. aculeate extracts and leaves were also found to be rich in carot-enoids and phenolic compounds, valuable phytochemicals which provide healing, an-ti-inflammatory, antifungal, and antioxidant activities, as well as analgesic potential. [3-7]. However, the characterization of the compounds present in P. aculeate leaves is scarce [8]. Moreover, it is crucial to understand the impact of extraction techniques on the proportion of bioactive compounds found in the extracts and leaves.”
  • Reviewer 3: Why the authors chose the ultrasound based extraction?
    Thank you for raising this point. This information was given in the manuscript as follows:
    “Given that the methods employed for obtaining the extracts can directly affect their antioxidant and biological potential [11], the ultrasound extraction technique is a prom-ising alternative. Compared to conventional methods for bioactive compound extraction, the ultrasonic method offers significant benefits regarding extraction time and process security [12]. The ultrasound-assisted extraction is also considered an environmentally friendly method because of its shorter extraction time and solvent consumption compared to conventional techniques [12]. Furthermore, the ultrasound-assisted extraction enhances the phytomolecules yields and extracts/isolated biological efficiency [11, 12].”
  • Reviewer 3: Literature have indicated that the ultrasound assisted extraction increases the yield of phytomolecules and thereby increase the biological efficacy of the isolate/ extract. Authors need to include a detailed account on the methodology and significance of ultrasound based method of extraction. Authors may refer and include the following articles in the paragraph;
    The authors thank the reviewer’s suggestion. As suggested, we added more detailed information regarding the chosen method.
  • Reviewer 3: The line "Furthermore, studies show that ora-pro-nóbis extracts and leaves are also rich in carotenoids and phenolic compounds, providing healing, anti-inflammatory and antioxidant activities, as well as analgesic potential and antifungal reaction [3-7]" needs to be expand with individual literature support. Authors can get more biological activities of the plant: Thanks for the recommendation. In fact, the references 3-7 collects information regarding studies about Pereskia aculeata Miller. Thus, the authors did not see the need to discuss them individually. Nevertheless, throughout the manuscript, more information regarding P. aculeata leaves benefitial activities and applications are presented, including their relation with the evaluated compounds.
  • Reviewer 3: Overall, the introduction needs to be re-written completely.  As recommended, the introduction section was rewritten covering the points raised by the reviewer.
  • Reviewer 3: Authors need to incorporate the voucher specimen number of the plant and identification details
    Due to the deadline for submitting the revised article, we were unable to obtain the voucher number for the species because this process takes longer, as it is carried out in another department of the University.
  • Reviewer 3: As I suggested earlier, the common name of the plant is relevant to the country only. So replace the same with scientific name or abbreviation:  We appreciate the reviewer’s suggestion. As one may notice, the suggested change was considered in this manuscript’s revised version.
  • Reviewer 3: The title emphasized "Ultrasonic extraction", hence, the details must be given including minute points about the frequency and set of instrument: As suggested, besides the details regarding extraction conditions previously mentioned, this new manuscript version presents the requested information. Thus information regarding requency and power of the instruments were included as follows:
    The extracts were prepared using a digital ultrasonic washer (SoniClean 2, Sanders Medical, Santa Rita do Sapucaí, Brazil) at room temperature (20 °C), frequency 42 kHz, and power 160 W. Four different extraction times were evaluated, namely, 10, 20, 30, and 40 min.”
  • Reviewer 3: While mentioning "Gram- positive or Gram- negative" the "G" must be in upper cases: The spelling was revised as suggested by the reviewer.
  • Reviewer 3: While mentioning instruments, the model, make, city and country details must be included: Thank you for raising this point. The requested information were included in the revised manuscript.
  • Reviewer 3: The table 3 must be formatted to indicate the MIC value only: Thank you for the observation. As suggested, MIC values were added in a revised version of Table 3.
  • Reviewer 3: Why the authors not estimated the zone of inhibition by disc diffusion assay; it needs to be done and result is to be included:
    We appreciate the reviewer’s observation. Although disc-diffusion assays are one of the standards for antimicrobial susceptibility testing, the microbroth dilution assay is considered the standard by the Clinical and Laboratory Standards Institute (CLSI). Thus, it can also provide multiple replicates and dilutions in a single plate. Additionally, because of the strong polar profile of some compounds, interactions with cellulose on the paper disk could occur and interfere with the inhibitory activity.
  • Reviewer 3: The selected strains (3 numbers) seems to be insufficient. Authors include data on one or two more strains: Thank you for the suggestion. Indeed, S. aureus and E. coli are considered representative standards for Gram-positve and Gram-negative strains. The E. coli used is also a beta-lactamase producing strain, thus representing a multidrug resistant microbe. Additionally, the MRSA also complements the assay as one of the more resistant strain. Moreover, these strains are recommended by CLSI for screening trials and are considered two of the most clinically relevant drug resistant bacteria by the US CDC (2019). Additionally, the use of ATCC strains is recommend for future comparison purposes for other studies.
  • Reviewer 3: The discussion contains limited literature support. I feel combining results and discussion makes it difficult to include a detailed analysis of the observed information and comparison with the available literature. Hence, I suggest to separate these two sections as stand alone sections: We appreciate the suggestion. The reviewer may notice that the data presented in the manuscript were discussed using at least one relevant literature reference. Moreover, a complete revision of the manuscript and English language were made for better comprehension of the information presented.
  • Reviewer 3: The formating of reference needs to be cross checked; there are some errors in the reference section (formating and punctuation errors): Thank you for the observation. As suggested the reference formating was revised.

Round 2

Reviewer 1 Report

The manuscript has been thoroughly reviewed, readability and presentation have been nicely improved. It can now be published, I just recommend the authors to check the text for minor oversights, such as missed superscripts (lines 130, 214, 219, 221, 281, 289, 304, 306, 316, 345, 346, 348, 349, 351, 355, 359, 361).

Author Response

Reviewer #1:

  • The manuscript has been thoroughly reviewed, readability and presentation have been nicely improved. It can now be published, I just recommend the authors to check the text for minor oversights, such as missed superscripts (lines 130, 214, 219, 221, 281, 289, 304, 306, 316, 345, 346, 348, 349, 351, 355, 359, 361).

Response: All superscripts have been revised and duly corrected. All suggestions were accepted and the entire manuscript was carefully revised. Thank you once again for your valuable comments. We appreciate the time and effort spent in this reviewing process.

Reviewer 2 Report

The manuscript improvements are sufficient for acceptance

Author Response

Reviewer #2:

The manuscript improvements are sufficient for acceptance.

Thank you once again for your valuable comments. We appreciate the time and effort spent in this reviewing process.

Reviewer 3 Report

Authors tried to improve the manuscript, however, the writing of the manuscript seems to be very premitive. Authors are expected to go beyond the reading of reviewer, however, in this case authors selected a couple of literature from the reviewers suggestion and concluded. This is not the best practise in scientific writing; writing a manuscript is equally important as experimentation and analysis. Overall, the changes made in the manuscript is minimal and it is not enough for publication in a quality journal like "Metabolites". Therefore I suggest a major revision once again and if it is found unsatisfactory, the article shall be rejected with decline to resubmit.

I also recommend to highlight all the changes the authors made

1. Abstract still lacks quantitative data and description of chemical constituents are general in nature (examples).

"such as organic and phenolic acids, flavonoids, and others"

"PS-MS proved to be a valuable technique to obtain the P. aculeate leaves extract chemical profile" what are the major findings?

"the first study to evaluate different times for ultrasound extraction of P. aculeate leaves"

(FTIR) identified carboxyl functional groups and proteins "which region of FTIR?

What is the finding of the study? it should be the concluding remarks of abstract and needs to be specific

2. The description on ultrasound method of extraction contain 2 references. I think reviewer went through the literature than the authors did. It should be detailed, because the entire study depends on it. I suggested several examples; the authors are expected to discuss more than that, however, they discussed a couple of them and concluded that ultrasound assisted method is the best. If the authors fails to describe the significance of the method of extraction in writing I suggest to carry out a comparative study with other extraction method and prove that the ultrasound is better for this particular plant.

3. I suggest to include the authentication details of the plant

4. IC50 values of the antioxidant activity must be clearly mentioned

5. Figure 1 quality is still low. very difficult to read axis details. Where is the chromatogram of plant. It should be in figure 1

6. I also suspect unwanted self citations in 35, 50, 70 etc. Authors need to remove them or explain the significance of these references.

Author Response

Reviewer #3:

- Authors tried to improve the manuscript, however, the writing of the manuscript seems to be very premitive. Authors are expected to go beyond the reading of reviewer, however, in this case authors selected a couple of literature from the reviewers suggestion and concluded. This is not the best practise in scientific writing; writing a manuscript is equally important as experimentation and analysis. Overall, the changes made in the manuscript is minimal and it is not enough for publication in a quality journal like "Metabolites". Therefore I suggest a major revision once again and if it is found unsatisfactory, the article shall be rejected with decline to resubmit.

Response: In order to meet all your considerations, the manuscript was completely revised. The introduction has been rewritten and added reader-relevant information and comparisons about Pereskia aculeate Miller and Ultrasound Assisted Extraction. Low resolution figures (1 and 2) were replaced and the discussion in Section 3.1 "Total phenolic compounds and antioxidant activity" was improved by discussing and comparing the results with other works in the field. Quantitative data has been added to the abstract.

- I also recommend to highlight all the changes the authors made.

Response: All changes made to the manuscript were highlighted in yellow.

1) Abstract still lacks quantitative data and description of chemical constituents are general in nature (examples).

"such as organic and phenolic acids, flavonoids, and others"

"PS-MS proved to be a valuable technique to obtain the P. aculeate leaves extract chemical profile" what are the major findings?

"the first study to evaluate different times for ultrasound extraction of P. aculeate leaves"

(FTIR) identified carboxyl functional groups and proteins "which region of FTIR?

What is the finding of the study? it should be the concluding remarks of abstract and needs to be specific.

Response: Lines 25 - 49. All requested quantitative data and chemical constituents were included in the abstract.

2) The description on ultrasound method of extraction contain 2 references. I think reviewer went through the literature than the authors did. It should be detailed, because the entire study depends on it. I suggested several examples; the authors are expected to discuss more than that, however, they discussed a couple of them and concluded that ultrasound assisted method is the best. If the authors fails to describe the significance of the method of extraction in writing I suggest to carry out a comparative study with other extraction method and prove that the ultrasound is better for this particular plant.

Response: Lines 53 – 120: The description of the method in the Introduction section has been improved and references have been added. Conventional extraction methods were also described, presenting the mechanism of ultrasound-assisted extraction and its advantages over traditional techniques. Some studies demonstrating the efficiency of UAE for obtaining bioactive compounds from plant matrices were also cited, as well as the reason for choosing this extraction technique was clearly presented.

3) I suggest to include the authentication details of the plant

Response: Thank you for suggestion. The department responsible for plants identification in our university requests a longer period (from 30 to 90 days) for validating the its authentication, as well as providing all details. In this sense, and considering the urgency for this answer and the publishing process, we will not be able to send this information at this time, such as the vouch number. We apologize for this inconvenience and we would appreciate your consideration to this matter. 

4) IC50 values of the antioxidant activity must be clearly mentioned

Response: Lines 242 - 247. The three methods (ABTS, FRAP and DPPH) used in this work to evaluate the antioxidant capacity of P. aculeate extracts do not express the result in IC50, but in the concentrations indicated in Table 1. The works (Cruz et al., 2021; Souza et al., 2016; Torres et al., 2022, 2021) used to compare the results with this manuscript, also did not perform analyzes based on mean inhibitory concentration (IC50).

5) Figure 1 quality is still low. very difficult to read axis details. Where is the chromatogram of plant. It should be in figure 1.

Response: Lines 319 – 324 and  418. The quality of Figure1 and 2 has been improved. Thanks for suggestion. A chromatographic analysis of the plants and its parts must be performed after an extraction process, which was made in this study highlighted by the different extracts obtained. As this analyses is not possible to me made with the whole plant, the extraction step allows us to obtain each fraction to be analyzed. Figures 1 and 2 were improved in quality, presenting the chromatograms of leaves extracts and the spectrum of the compounds. In addition, Figure 4 also presents the spectrogram of leaves extracts.

6) I also suspect unwanted self citations in 35, 50, 70 etc. Authors need to remove them or explain the significance of these references.

Response: Reference 27 was used to report the methodology in “Section 2.10. Chemical profile determination by Paper Spray Mass Spectrometry (PS—MS)”. Reference 28 contributes to the discussion in “Section 3.6. Chemical profile by paper spray mass spectrometry”. Reference 80 contributed to the discussion in “Section 3.6. Chemical profile by paper spray mass spectrometry”. We verified that the reference was out of contexto. Correia, V.T. da V.; D’Angelis, D.F.; Macedo, M.C.C.; Ramos, A.L.C.C.; Vieira, A.L.S.; Queiroz, V.A.V.; Augusti, R.; Ferreira, A.A.; Fante, C.A.; Melo, J.O.F. Perfil Químico Da Farinha Extrusada de Sorgo Do Genótipo BRS 305 Por Paper Spray. Res. Soc. Dev. 2021, 10, e40710111414, doi:10.33448/rsd-v10i1.11414” and has been removed.

The entire manuscript was carefully revised. Thank you once again for your valuable comments. We appreciate the time and effort spent in this reviewing process.

References

Cruz, T.M., Santos, J.S., do Carmo, M.A.V., Hellström, J., Pihlava, J.M., Azevedo, L., Granato, D., Marques, M.B., 2021. Extraction optimization of bioactive compounds from ora-pro-nobis (Pereskia aculeata Miller) leaves and their in vitro antioxidant and antihemolytic activities. Food Chem. 361. https://doi.org/10.1016/j.foodchem.2021.130078

Souza, L.F., Caputo, L., De Barros, I.B.I., Fratianni, F., Nazzaro, F., De Feo, V., 2016. Pereskia aculeata muller (Cactaceae) leaves: Chemical composition and biological activities. Int. J. Mol. Sci. 17, 1–12. https://doi.org/10.3390/ijms17091478

Torres, T.M.S., Álvarez-Rivera, G., Mazzutti, S., Sánchez-Martínez, J.D., Cifuentes, A., Ibáñez, E., Ferreira, S.R.S., 2021. Neuroprotective potential of extracts from leaves of ora-pro-nobis (Pereskia aculeata) recovered by clean compressed fluids. J. Supercrit. Fluids 179. https://doi.org/10.1016/j.supflu.2021.105390

Torres, T.M.S., Guedes, J.A.C., de Brito, E.S., Mazzutti, S., Ferreira, S.R.S., 2022. High-pressure biorefining of ora-pro-nobis (Pereskia aculeata). J. Supercrit. Fluids 181, 105514. https://doi.org/10.1016/j.supflu.2021.105514

Round 3

Reviewer 3 Report

No more comments